# Resonant X-ray excitation of the nuclear clock isomer $^{45}$Sc

Yuri Shvyd'ko[1✉], Ralf Röhlsberger[2,3,4,5], Olga Kocharovskaya[6], Jörg Evers[7], Gianluca Aldo Geloni[8], Peifan Liu[1], Deming Shu[1], Antonino Miceli[1], Brandon Stone[1], Willi Hippler[2,3], Berit Marx-Glowna[2,3], Ingo Uschmann[4], Robert Loetzsch[4], Olaf Leupold[5], Hans-Christian Wille[5], Ilya Sergeev[5], Miriam Gerharz[7], Xiwen Zhang[6], Christian Grech[5], Marc Guetg[5], Vitali Kocharyan[5], Naresh Kujala[8], Shan Liu[5], Weilun Qin[5], Alexey Zozulya[8], Jörg Hallmann[8], Ulrike Boesenberg[8], Wonhyuk Jo[8], Johannes Möller[8], Angel Rodriguez-Fernandez[8], Mohamed Youssef[8], Anders Madsen[8] & Tomasz Kolodziej[9]

Resonant oscillators with stable frequencies and large quality factors help us to keep track of time with high precision. Examples range from quartz crystal oscillators in wristwatches to atomic oscillators in atomic clocks, which are, at present, our most precise time measurement devices[1]. The search for more stable and convenient reference oscillators is continuing[2–6]. Nuclear oscillators are better than atomic oscillators because of their naturally higher quality factors and higher resilience against external perturbations[7–9]. One of the most promising cases is an ultra-narrow nuclear resonance transition in $^{45}$Sc between the ground state and the 12.4-keV isomeric state with a long lifetime of 0.47 s (ref. 10). The scientific potential of $^{45}$Sc was realized long ago, but applications require $^{45}$Sc resonant excitation, which in turn requires accelerator-driven, high-brightness X-ray sources[11] that have become available only recently. Here we report on resonant X-ray excitation of the $^{45}$Sc isomeric state by irradiation of Sc-metal foil with 12.4-keV photon pulses from a state-of-the-art X-ray free-electron laser and subsequent detection of nuclear decay products. Simultaneously, the transition energy was determined as $12{,}389.59^{+0.15(\text{stat})}_{+0.12(\text{syst})}$ eV with an uncertainty that is two orders of magnitude smaller than the previously known values. These advancements enable the application of this isomer in extreme metrology, nuclear clock technology, ultra-high-precision spectroscopy and similar applications.

The scientific potential of the $^{45}$Sc resonance, together with the possibility of its resonant excitation by photons from modern accelerator-based sources of hard X-rays (no radioactive parent isotope is available for $^{45}$Sc), was identified more than 30 years ago[11]. However, earlier attempts to induce resonant excitation at third-generation synchrotron radiation sources were not successful, mainly because of the lack of sufficient spectral flux. This flux constraint was overcome only recently with the advent of narrow-band XFELs working at a high repetition rate, such as the European XFEL (EuXFEL)[12,13].

Long-lived nuclear isomeric states with resonance quality factors comparable to or exceeding those used in modern atomic optical clocks[1] have the potential for revolutionizing quantum metrology, clock technology, chronometric geodesy and gravimetry and enable fundamental physics tests that rely on the measurement of time or frequency with utmost precision, such as the search for time–space variation of the fundamental constants, dark matter, violation of the Lorentz invariance and Einstein's equivalence principle[7–9,14–17]. These isomeric states could also lead to the improvement of resolution of the coherent nuclear forward scattering spectroscopy by

orders of magnitude[11] and to compact long-lived nuclear quantum memory[18].

The main advantage of nuclear transitions over atomic transitions in these applications is their lower sensitivity to perturbations caused by electric and magnetic fields because of the tiny size of nuclei and small magnitudes of nuclear electromagnetic moments. This could lead to higher accuracy of the nuclear clocks and could make it possible to use macroscopic ensembles of nuclear oscillators in bulk solids. Using large ensembles, along with a higher transition frequency, would yield stability that is orders of magnitude higher. Moreover, ultra-low temperatures would not be required because of the Mössbauer effect that suppresses thermal line broadening. For these reasons, the concept of a nuclear clock, especially of a Mössbauer nuclear clock, has recently drawn considerable attention[2–4,6].

Most long-lived nuclear isomeric states have resonance energies above 30 keV (fig. 1 of ref. 2) and hence cannot be resonantly driven by the existing sources of coherent radiation. Among them is a 88-keV transition in $^{109}$Ag with a 57-s lifetime that was investigated with radioactive sources for almost 60 years[15,19] without yielding conclusive

[1]Argonne National Laboratory, Lemont, IL, USA. [2]Helmholtz Institute Jena, Jena, Germany. [3]GSI Helmholtzzentrum für Schwerionenforschung, Darmstadt, Germany. [4]Friedrich-Schiller-Universität Jena, Jena, Germany. [5]Deutsches Elektronen-Synchrotron (DESY), Hamburg, Germany. [6]Texas A&M University, College Station, TX, USA. [7]Max Planck Institute for Nuclear Physics, Heidelberg, Germany. [8]European X-Ray Free-Electron Laser Facility, Schenefeld, Germany. [9]National Synchrotron Radiation Centre SOLARIS, Kraków, Poland. ✉e-mail: shvydko@anl.gov

**Table 1 | Parameters of $^{45}$Sc, taken from or based on ref. 32 unless otherwise stated, with updated values measured or used in this work highlighted in bold**

| Parameter | Notation | Magnitude | References |
|---|---|---|---|
| Ground state | | | |
| Spin and parity | $I_g^\pi$ | $7/2^-$ | |
| Magnetic moment | $\mu_g$ | $+4.756487(2)\,\mu_N$ | |
| Quadrupole moment | $Q_g$ | $-0.22(1)\,b$ | |
| Isotopic abundance | $\eta$ | 100% | |
| Excited state | | | |
| Energy | $E_0$ | 12,400(50) eV | 32,49,50 |
| | | **12,389.59$^{\pm0.15(stat)}_{+0.12(syst)}$ eV** | |
| Lifetime | $\tau_0$ | 470(6) ms | 10,32,51 |
| Natural linewidth | $\Gamma_0 = \hbar/\tau_0$ | 1.40(2) feV | |
| Quality factor | $Q = E_0/\Gamma_0$ | $8.85(10)\times10^{18}$ | |
| Multipolarity | | M2 | |
| Spin and parity | $I_e^\pi$ | $3/2^+$ | |
| Magnetic moment | $\mu_e$ | $+0.368(5)\,\mu_N$ | 52 |
| Quadrupole moment | $Q_e$ | $+0.318(22)\,b$ | 52 |
| Internal conversion coefficients | $\alpha, \alpha_K$ | 632(71), 474 | 32,51,53 |
| | | **424, 363** | 38 |
| Cross-section | $\sigma_0$ in equation (3) | $1.26(15)\times10^{-20}\,cm^2$ | |
| | | **$1.9(5)\times10^{-20}\,cm^2$** | |

Wavelength corresponding to measured $^{45}$Sc energy: $\lambda_0 = 1.00071$ Å.

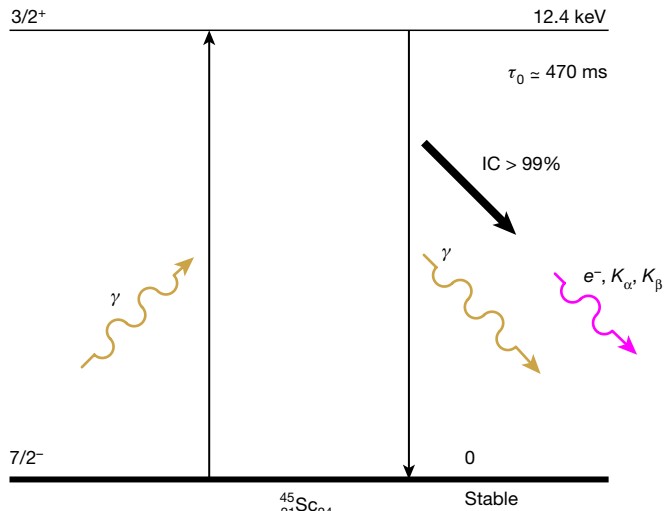

**Fig. 1 | Excitation and decay scheme of the 12.4-keV first excited state of the $^{45}$Sc nucleus.** The nonradiative internal conversion decay channel dominates, resulting in the emission of Sc characteristic X-ray fluorescent photons and Auger electrons. $K_\alpha$ (4.09 keV) and $K_\beta$ (4.46 keV) fluorescent photons were detected in this study.

about ±50 eV (ref. 32), was determined with more than 100 times higher accuracy as 12,389.59$^{\pm0.15(stat)}_{+0.12(syst)}$ eV. Furthermore, we obtained estimates for the resonance cross-section and for the coefficients of internal conversion of this transition. The high average spectral flux of the incident X-rays and the very low background noise in the detection of the nuclear decay products were crucial for the success of the experiment.

## Approach

X-rays generated by EuXFEL were used to excite $^{45}$Sc nuclei in a solid-state target from the ground into the isomeric state. The pulsed source provided 12.4-keV X-rays with the highest spectral density and a pulse duration much less than the isomer lifetime $\tau_0 = 0.47$ s. Nuclear decay products were detected with a characteristic delay of about $\tau_0$ after the excitation. To achieve an utmost reduction of detection background after irradiation, the detectors were offset from the beam path, and the target was mechanically moved out of the X-ray beam into this offset position after each pulse train to record the delayed fluorescence signal. Two low-noise Si solid-state drift detectors were used to detect nuclear decay (Fig. 2 and section 'Experimental challenges').

As $^{45}$Sc exhibits a rather high coefficient of internal conversion $\alpha$, in the range of about 400–650 (that is, the probability of decay via internal conversion greatly exceeds the probability of a radiative decay; Table 1), we used this incoherent nuclear decay channel for resonance detection. Nuclear decay via internal conversion results in the ejection of an electron from an inner shell (predominantly $1s$, that is, K shell). The resulting core hole is filled by the emission of characteristic $K_\alpha$ (4.09 keV) and $K_\beta$ (4.46 keV) X-rays, corresponding to the $2p$–$1s$ and $3p$–$1s$ transitions, respectively, and/or emission of Auger electrons. Although Auger electrons dominate (the K-shell fluorescence yield is only $\omega_K = 0.19$ for Sc)[33], we chose to detect X-ray fluorescence, as X-ray detectors (unlike electron detectors) do not require a special environment. Moreover, along with the 4-keV photons, they can simultaneously detect 12.4-keV forward-scattered photons with a high spectral resolution. It is expected that this set-up will produce elastically and coherently scattered delayed 12.4-keV photons in the forward direction, a phenomenon known as nuclear-resonant forward scattering (NFS)[34–36].

evidence yet for resonance detection. Other approaches have used bremsstrahlung to populate indirectly long-lived isomeric states such as the 39-keV level of $^{103}$Rh using higher-energy broadband excited states[20]. Fortunately, some lower-energy isomers exist that qualify for coherent resonant electromagnetic excitation. The most prominent is $^{229}$Th, which has an extraordinarily low transition energy of about 8 eV (refs. 2,4–9). This transition is especially attractive because it is within the reach of ultraviolet light frequency combs. Notable progress was achieved recently in a more precise determination of its energy[4–6]. Still, exciting this resonance with photons remains an open challenge[2–9]. An alternative, although indirect, route to populate the isomeric state could proceed using X-ray excitation of the 29.2-keV broadband second excited state of $^{229}$Th (ref. 3).

Now, with the availability of modern X-ray free-electron lasers (XFELs)[12,13,21–25], several lower-energy and long-lived nuclear isomeric resonances are within reach. One of the most promising is $^{45}$Sc, which has a transition energy of $E_0 = 12.4$ keV, an isomer lifetime of $\tau_0 = 0.47$ s and a natural linewidth of $\Gamma_0 = \hbar/\tau_0 = 1.4$ feV, resulting in an extremely high quality factor of $Q = E_0/\Gamma_0 \simeq 10^{19}$ (see Table 1 for the nuclear parameters of $^{45}$Sc and Fig. 1 for a schematic of the nuclear transitions). This quality factor is six orders of magnitude larger than that of the 14.4-keV resonance of $^{57}$Fe—a workhorse of Mössbauer spectroscopy[26–28]—and surpasses other measurable Mössbauer resonances by orders of magnitude[29,30]. A high-quality resonance of this kind would enable the measurement of gravitational red shift or gravitational time dilation at displacements much shorter than the 1-mm record established on the basis of a Sr atomic clock[31]. Furthermore, $^{45}$Sc is a stable isotope with 100% natural abundance and is readily available either as ultra-pure Sc metal or as $Sc_2O_3$, in which its 12.4-keV transition has a high Lamb–Mössbauer factor $f_{LM} \simeq 0.75$ at room temperature. All of these facts make $^{45}$Sc superior to any other candidate for a Mössbauer nuclear clock.

Here we report on the resonant excitation of the long-lived ultra-narrow 12.4-keV nuclear state of $^{45}$Sc using XFEL pulses. In this experiment, the transition energy, previously known to an uncertainty of only

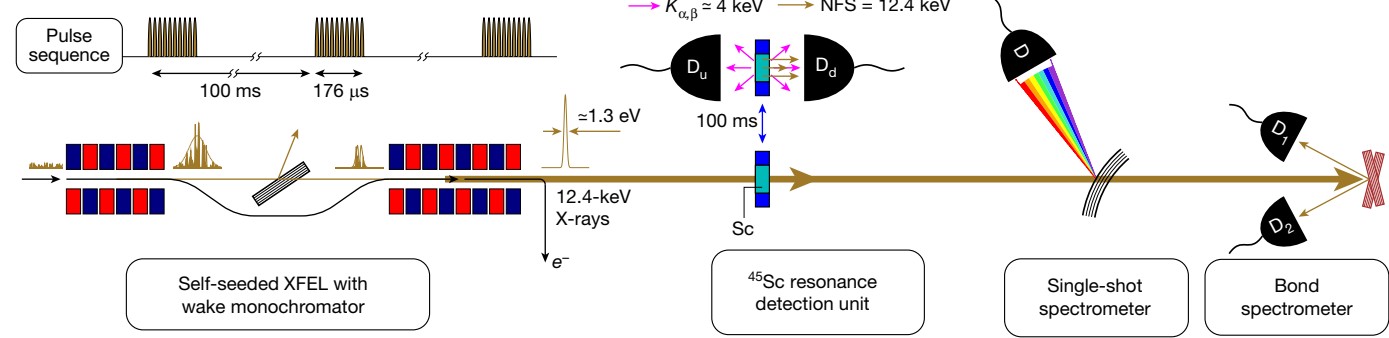

**Fig. 2 | Schematic of the experiment designed to resonantly excite $^{45}$Sc nuclei from the ground state to the long-lived ultra-narrow 12.4-keV excited state using XFEL pulses and to detect the $^{45}$Sc resonance.** Upstream and downstream X-ray counters $D_u$ and $D_d$ detect time-delayed nuclear decay products–both inelastic $K_{\alpha,\beta}$ fluorescence and coherent elastic NFS. To minimize the detection background, the decay product detectors are offset from the beam path and the sample is moved to the detectors after irradiation with each pulse train. The resonance energy is measured with X-ray single-shot and Bond spectrometers.

Observation of such photons would not only enable detection of the $^{45}$Sc resonant excitation but also–if the time dependence of the NFS could be measured–provide information on the resonance linewidth[11]. However, the probability of detecting 12.4-keV photons was expected to be significantly lower than that of 4-keV photons, because of both the high internal conversion coefficient and the anticipated large yet unknown broadening of the resonance line. Therefore, we focused on the detection of 4-keV photons.

Because the energy of the 12.4-keV state was known before the experiment to an accuracy of only ±50 eV (ref. 32), the energy of the incident X-rays had to be scanned over a comparatively large range of 100 eV and accurately measured in parallel with detecting the nuclear decay products. The relative photon energy was continuously measured with a single-shot spectrometer (SSS) and then related to the absolute energy scale using Bond spectrometer measurements (Fig. 2 and section 'Absolute resonance energy').

Although the procedure seems simple, many challenges had to be overcome to realize the experiment, as detailed in the section 'Experimental challenges'.

Initially, two energy scans were performed over a range of ±50 eV around 12.40 keV. After the candidate $^{45}$Sc resonance energy $E_0$ was located, the subsequent scans were constrained to a smaller range of ±5 eV around this value. The pure data acquisition time was around 30 h, during which about $10^{20}$ photons from the EuXFEL were directed to the $^{45}$Sc targets. In total, about 93 nuclear decay events were detected. Although the number of the nuclear decay events detected was relatively small (about 93), the number of $^{45}$Sc nuclei excited in the experiment was estimated to be much larger around $7 \times 10^3$ (Methods). Because of the extremely low detector background, this number was sufficient to reveal the $^{45}$Sc resonance and determine the resonance energy with high accuracy.

### Resonant excitation of $^{45}$Sc

Figure 3 shows the two-dimensional plots of counts from the upstream and downstream decay detectors, $D_u$ and $D_d$, respectively, with a plot of all detected photons with energies $E_f \lesssim 26$ keV at all incoming photon energies $E_i$ (Fig. 3a), and close-ups of regions of interest (ROIs) around the energies of the fluorescence and NFS photons (Fig. 3b,c). In Fig. 3a, the vertical red lines indicate the constrained scan range of ±5 eV around the $^{45}$Sc resonance energy $E_0$. Because of this scan-range reduction, the density of the detected (background) photons seems to be larger between the vertical red lines.

Figure 3b shows a close-up of the 4.3-keV ROI in the reduced incident energy $E_i$ scan range. It shows two clusters of about 93 counts that are centred at detector energy values $E_f$ that correspond to the energies of

Sc $K_\alpha$ (4.09 keV) and $K_\beta$ (4.46 keV) fluorescence. Because these events were recorded with at least a 20-ms delay after the arrival of the XFEL pulses on the $^{45}$Sc target, this observation presents clear evidence of the detection of decay products of the long-lived 12.4-keV excited state of $^{45}$Sc nuclei resonantly excited by XFEL pulses.

Figure 3c shows a close-up of the 12.4-keV ROI in the same $E_i$ range. There are just a couple of counts at these energies. Thus, as anticipated, an NFS signal could not be detected in this experiment (see section 'Nuclear forward scattering').

### Resonance energy

Figure 4 shows the data of Fig. 3b but integrated over $E_f$ in the detection ROI. The graph shows all delayed Sc K-shell fluorescence counts as a function of the energy of the incoming X-ray photons. A clear resonance peak with a high signal-to-noise ratio of around 65 is observed, centred at $E_0$. The measurements performed with the Bond-type X-ray spectrometer[37] (Fig. 2 and section 'Absolute resonance energy') enabled us to relate the relative to the absolute energy scale with high accuracy and determine the energy of $^{45}$Sc resonance as $E_0 = 12,389.59^{\pm0.15(stat)}_{+0.12(syst)}$ eV. We note that the peak width is 1.32(12) eV, reflecting the energy width of the XFEL pulses.

### Count rates, internal conversion coefficients and resonance cross-section

The scale on the left of Fig. 4 presents normalized counts: that is, the integral over the spectrum equals the total number of the delayed K-shell fluorescence photons (approximately 93) recorded in the experiment.

The scale on the right presents the nuclear-resonance-assisted K-shell fluorescence yield that would be measured for one incident resonant photon (in the natural linewidth $\Gamma_0$) on the target. The peak yield derived from the experimental data is $\Sigma_K^{exp} = 0.44(10)$ ph$_K$/ph$_{\Gamma_0}$ (for details, see section 'Incoherent scattering count rates'). However, the theory outlined in that section predicts a peak yield of $\Sigma_K = 0.23$ ph$_K$/ph$_{\Gamma_0}$ –smaller by a factor of 1.9(4)–when the internal conversion coefficients $\alpha$ and $\alpha_K$ adapted from ref. 32 (Table 1) are used in the calculations.

A better agreement between the theoretical and experimental peak yields is achieved if smaller values of $\alpha$ and $\alpha_K$ are used. In particular, using the internal conversion coefficients predicted by the state-of-the-art internal conversion theory (https://bricc.anu.edu.au/)[38], the nuclear-resonance-assisted K-shell fluorescence yield becomes $\Sigma_K = 0.39$ ph$_K$/ph$_{\Gamma_0}$, which is in good agreement with the experiment. The updated $\alpha$ and $\alpha_K$ values and a nuclear resonance cross-section $\sigma_0$ calculated using these internal conversion coefficients are shown in bold in Table 1.

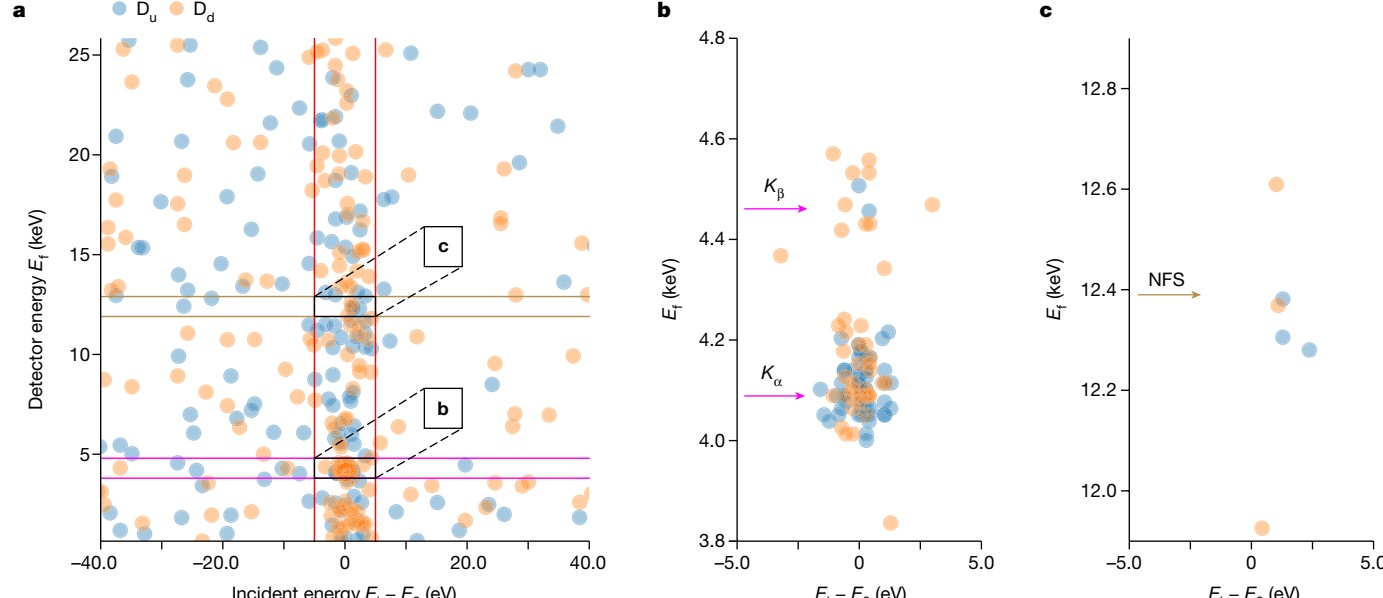

**Fig. 3 | Counts from the $D_u$ and $D_d$ detectors plotted as the energy $E_f$ of the detected X-ray photons versus the incident X-ray photon energy (expressed as the difference of the incident energy $E_i$ and the resonance energy $E_0$).** The photons were recorded in a time window of about 20–80 ms after every pulse-train excitation. Each photon is shown by a blue circle for $D_u$ and a yellow circle for $D_d$. **a**, All detector counts with $E_f \lesssim 26$ keV for incident energies in a range of $-40$ eV $\lesssim E_i - E_0 \lesssim 40$ eV around the $^{45}$Sc resonance energy $E_0$. Vertical red lines indicate a 10-eV ROI around $E_i = E_0$. Horizontal dark yellow lines indicate a 1-keV ROI around $E_f = 12.4$ keV, whereas the horizontal magenta lines indicate a 1-keV ROI around $E_f = 4.3$ keV, the approximate energy of the fluorescence photons. **b**, Close-up of the 4.3-keV ROI, showing two clusters of counts centred at the energies of Sc $K_\alpha$ and $K_\beta$ fluorescence. **c**, Close-up of the 12.4-keV ROI.

## Discussion

Taken together, the resonant excitation of the $^{45}$Sc resonance and the accurate measurement of the resonance energy reported here open new prospects in ultra-high-precision spectroscopy, nuclear clock technology and extreme metrology in the regime of hard X-rays. This work further demonstrates that the state-of-the-art high-repetition-rate narrow-band XFELs provide a promising experimental platform for the study of ultra-long-lived nuclear resonances at energies of hard X-rays. The improved accuracy of the resonance-energy determination sets the stage for the next step: an observation of the time dependence of the coherent NFS. This observation would not only provide the actual

spectral width of the $^{45}$Sc resonance[11] but also immediately enable important applications in extreme metrology and ultra-high-precision coherent NFS spectroscopy, because the narrowness of the $^{45}$Sc resonance is expected to surpass all currently accessible Mössbauer resonances by orders of magnitude. An example could be an interferometric measurement of gravitational red shift in the time domain, which could potentially reduce the required height difference to the submillimetre range (compared with 22.6 m in the historic Pound–Rebka experiment using $^{57}$Fe[39]). Coherent NFS from $^{45}$Sc could also be used for measuring extremely weak couplings in solids that result in energy shifts or splitting much below the currently accessible 1-MHz range.

The observation of $^{45}$Sc NFS will require extra steps such as cooling the nuclear target to exclude the thermal line broadening, minimizing or suppressing inhomogeneous broadening and maximizing the NFS signal by increasing the target thickness (see section 'Nuclear forward scattering').

We foresee the development of a $^{45}$Sc-based nuclear clock in the future. Achieving this goal will require a further increase in the resonant spectral flux using improved narrow-band 12.4-keV X-ray sources and frequency combs stretching up to this energy. There are two possible routes to this end. The first route is based on using an X-ray free-electron-laser oscillator (XFELO)[40] and a hard X-ray comb generated by a nuclear-resonance-stabilized XFELO[41]. The realization of such devices is presently in progress[42,43]. Specifications for the X-ray source and for the nuclear clock procedure conceptually outlined in ref. 41 will become more explicit after the observation of $^{45}$Sc NFS and optimization of the $^{45}$Sc target for maximum NFS signal strength and the smallest linewidth of the actual $^{45}$Sc resonance. The second route is to use intracavity high harmonic generation (HHG) as the frequency comb[44]. Recently, the cut-off photon energy in HHG has been pushed into the kiloelectron volt range, that is, to 5.2 keV (ref. 45), and the intracavity HHG was spread into the extreme ultraviolet light range[46], with an expected spectral width of each harmonic of the order of 0.1 Hz (refs. 47,48), which would perfectly match the spectral width of the $^{45}$Sc resonance.

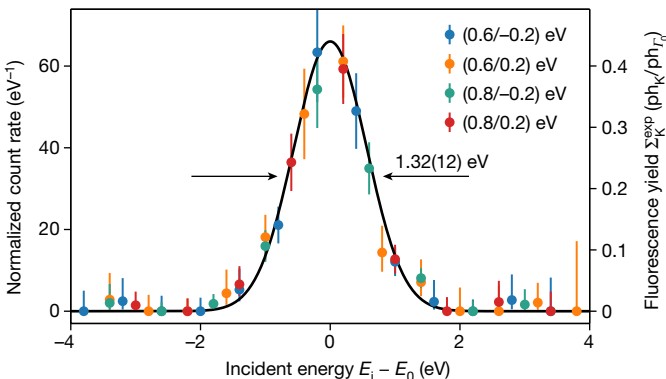

**Fig. 4 | Time-delayed Sc K-shell fluorescence recorded by detectors $D_u$ and $D_d$ as a function of incoming X-ray photon energy (expressed as the difference of the incident energy $E_i$ and the resonance energy $E_0$).** The solid black line shows the resonance line obtained from the data evaluation (see section 'Data analysis'). The coloured dots show exemplary binned data, with different incident energy bin sizes and shifts of the bin grid. The spectral width of 1.32(12) eV reflects the spectral width of the incoming XFEL radiation.

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

## Methods

### Experimental challenges

Although the experimental approach seems simple, many challenges had to be overcome to use our measurement and detection scheme at an XFEL facility, as explained below.

**Spectral flux.** An X-ray photon spectral flux of at least about 0.1 ph s$^{-1}$ $\Gamma_0^{-1} \simeq 0.7 \times 10^{14}$ ph s$^{-1}$ eV$^{-1}$ is required to detect the $^{45}$Sc resonance in a reasonable time. These spectral flux values are becoming possible with the advent of narrow-band self-seeded hard X-ray FELs[22,24,25,54,55] operating at a high repetition rate, resulting in up to three orders of magnitude higher average spectral flux compared with values at storage-ring-based sources of hard X-rays[56]. The EuXFEL recently became the first to our knowledge operational facility of this kind[12,13].

The experiment was performed at the Materials Imaging and Dynamics (MID) instrument[57] at the EuXFEL. Figure 2 shows the schematic of the experiment. The EuXFEL operated as a self-seeding XFEL—that is, X-rays from the first half of the XFEL undulator were used to seed the second half through a diamond-crystal-based (wake) monochromator[13,22,25,55,58,59].

The EuXFEL generated 12.4-keV X-ray pulses with an average rate of 4 kHz, an average pulse energy of about 350 μJ, a spectral width of around 1.3 eV (full width at half maximum), an average X-ray power of about 1.4 W, a photon flux of about $7.0 \times 10^{14}$ ph s$^{-1}$ and a spectral flux of about $5 \times 10^{14}$ ph s$^{-1}$ eV$^{-1}$ or about 0.7 ph s$^{-1}$ $\Gamma_0^{-1}$. The actual spectral flux transported to $^{45}$Sc targets was $F_{\Gamma_0} \simeq 0.31(6)$ ph$^{-1}$ s$^{-1}$ $\Gamma_0^{-1}$ because of attenuation on the way from the source to the target by a factor of $A = 0.44(9)$.

**Background suppression.** The low signal strength estimated for this experiment calls for a detector system with a noise floor not exceeding $10^{-3}$ ph s$^{-1}$ in the presence of the huge incoming photon flux. This noise floor requirement has been met by using temporal, energy, spatial and polarization degrees of freedom to discriminate detector signals against background counts.

The EuXFEL provides a unique time sequence of X-ray pulses, which is perfectly suited for the detection of the $^{45}$Sc resonance with a very good signal-to-noise ratio. X-rays arrive in submillisecond pulse trains having a periodicity of 100 ms (Fig. 2). In this experiment, each pulse train consisted of 400 X-ray pulses spaced by 440 ns, with each pulse having a duration of a few tens of femtoseconds. The resulting pulse-train duration of 176 μs is much shorter than the natural lifetime $\tau_0$ of $^{45}$Sc. This pulse-train sequence is ideal for efficient excitation and detection of the $^{45}$Sc resonance, because it enables temporal discrimination of detector signals against a huge incident flux during the excitation process: during the dark time between the pulse trains, the millisecond-delayed $^{45}$Sc resonance decay products can be detected in near-complete absence of background signals.

Additional background reduction is achieved by using energy-resolving Si solid-state drift detectors (Amptek X123, labelled D$_u$ and D$_d$ in Fig. 2) with an energy resolution of 150 eV in the X-ray regime below 15 keV. This detector choice facilitates sharp energy discrimination of the 4-keV $K_{\alpha,\beta}$ characteristic fluorescent photons against all other X-ray energies.

Equally essential for background suppression is the relation between the irradiation position and the detection position. The detectors are placed at around 12-mm offset from the incident beam, and the $^{45}$Sc target is shifted swiftly back and forth from the beam path to a position between the detectors every 100 ms by a dedicated linear motion system (Fig. 2). The sample dwells at the irradiation position for around 10 ms and at the detector position for about 60 ms; the transit time between positions is approximately 15 ms. The periodic motion is synchronized with the arrival of the 12.4-keV X-ray pulse trains. The detector offset is in the horizontal plane to suppress X-ray Thomson scattering, because the electric vector of the incident photon polarization is also in the horizontal plane.

As a result of these measures, the total detector background count rate in the experiment was as low as about $2 \times 10^{-4}$ ph s$^{-1}$.

**The $^{45}$Sc target.** The choice of the $^{45}$Sc target was dictated by the necessity of achieving high nuclear resonance optical thickness, low absorption of the nuclear decay products, resilience to radiation damage, simplicity in fabrication and integration into the motion and detection system. Commercially available Sc-metal foil with a thickness $L = 25$ μm was chosen as a target; the foil has a number density $N = 3.98 \times 10^{22}$ cm$^{-3}$ of $^{45}$Sc nuclei and an area number density $\rho = NL = 1 \times 10^{20}$ cm$^{-2}$. With this choice, the nuclear resonance optical thickness had a substantial value of $\sigma_0 \rho = 1.87$ (see Table 1 for $\sigma_0$). The target thickness of 25 μm was chosen to maximize the collection efficiency of the 4-keV photons as it corresponds to one attenuation length ($L_K \simeq 23$ μm) of 4-keV photons in Sc metal.

**Radiation damage.** Every X-ray pulse train with an energy of about 65 mJ heats the 25-μm thick Sc-metal target by an estimated temperature of about 100 K. The heat conductivity of Sc metal is low—15.8 W m$^{-1}$ K$^{-1}$, comparable to that of stainless steel. Radiation damage to the target is thus inevitable. Special measures were taken to avoid damage, including water and air cooling, defocusing of the beam (the beam footprint was about 2 mm$^2$) and periodic replacement of the target. Two targets had to be replaced in the course of the experiment because of partial radiation damage.

**Uncertainty of the resonance energy.** Because of the ±50 eV uncertainty in the $^{45}$Sc resonance energy before this experiment, the EuXFEL photon energy had to be scanned over a rather large energy range around the anticipated 12.40-keV value while simultaneously detecting the nuclear decay products (EuXFEL currently delivers photons in a self-seeding mode in the energy range of 6–14.4 keV). Such scans require simultaneous and accurate measurement of the energy of the incident photons. An SSS[60–63] installed in the endstation of MID (Fig. 2) measures the spectral distribution and its peak photon energy in relative units (SSS scale). The peak value on the SSS energy scale is related to the absolute incident photon energy scale $E_i$ with an accuracy of ≲0.3 eV by measuring selected photon energies using a Bond-type spectrometer[37] (Fig. 2 and section 'Absolute resonance energy').

### Incoherent scattering count rates

The attenuation of monochromatic X-rays of a spectral flux $F_0$ propagating through a target of thickness $L$ consisting of resonant nuclei with a number density $N$ can be calculated as

$$F(E, L) = F_0 \exp[-\sigma(E)NL - L/L_0], \qquad (1)$$

where the energy dependence of the total resonance cross-section $\sigma(E)$ of both resonant elastic (radiative) scattering and inelastic (internal conversion) scattering is given by the Breit–Wigner formula:

$$\sigma(E) = \sigma_0 \frac{\Gamma \Gamma_0 / 4}{(E - E_0)^2 + (\Gamma/2)^2}, \quad \Gamma = \Gamma_0 + \Delta\Gamma, \qquad (2)$$

$$\sigma_0 = \frac{4\pi}{k^2} \frac{2I_e + 1}{2(2I_g + 1)} \frac{1}{1 + \alpha}. \qquad (3)$$

Here $\sigma_0$ is the total nuclear resonance cross-section; $E_0$ is the resonance energy; $\Gamma_0 = \Gamma_1 + \Gamma_2 = \Gamma_1(1 + \alpha)$ is the total resonance width, which comprises the elastic and inelastic widths $\Gamma_1$ and $\Gamma_2$, respectively; $\alpha$ is the internal conversion coefficient; $k = E_0/\hbar c$; $\hbar$ is Planck's constant; $c$ is the speed of light in vacuum; and $L_0$ is the photoabsorption length

of the incident X-rays in the target. The standard expression for the Breit–Wigner cross-section[64] is modified here to take into account an additional broadening $\Delta\Gamma$ to the natural linewidth of the resonance. For other notations, see Table 1.

Using equations (1)–(3), a total incoherent nuclear-resonant yield $\Sigma_0$ can be calculated. This yield is a relative number reflecting the contributions of all decay products incoherently scattered into $4\pi$ by the resonant nuclei in the target and normalized to resonant photon flux $F_0\Gamma_0$ (incident photon flux within $\Gamma_0$). It can be calculated as

$$\Sigma_0 = \frac{1}{F_0\Gamma_0} \int dE\ \sigma(E)N \int_0^L dx\ F(E,x). \tag{4}$$

If we assume that the resonant absorption length $L_R = [\sigma_0 N(\Gamma_0/\Gamma)]^{-1}$ becomes large compared with both $L$ and $L_0$ because of a large degree of line broadening (that is, $\Gamma \gg \Gamma_0$), equation (4) can be approximated as

$$\Sigma_0 = \frac{\pi}{2}\sigma_0 N L_0 [1 - \exp(-L/L_0)]. \tag{5}$$

This equation, however, ignores the possible losses of the nuclear decay products in the target, which will be addressed later.

A partial nuclear-resonance-assisted K-shell X-ray fluorescence yield measured in this experiment can be calculated as

$$\Sigma_K = \Sigma_0 \frac{\alpha_K \omega_K}{1+\alpha}. \tag{6}$$

Here $\alpha_K$ is a partial K-shell internal conversion coefficient ($\alpha = \alpha_K + \alpha_L + ...$) and $\omega_K$ is the K-shell X-ray fluorescence yield. In the case of Sc atoms, $\omega_K = 0.19(1)$; that is, the fluorescence yield is much smaller than the Auger yield[33,65].

In the particular case of the Sc-metal target of thickness $L = 25\ \mu m$ used in the experiment with number density $N = 3.98 \times 10^{22}\ cm^{-3}$ of $^{45}Sc$ nuclei in the Sc metal, $L_0 = 60\ \mu m$ and with values of $\sigma_0$ and $\alpha_K/(1+\alpha) \simeq 0.75$ (Table 1), we obtain $\Sigma_K = 0.23\ ph_K/ph_{\Gamma_0}$ for the anticipated partial nuclear-resonance-assisted K-shell fluorescence yield per incident photon within the natural linewidth. The target thickness $L$ was chosen in the experiment to be about one photoabsorption length of K-shell fluorescent photons ($L_K \simeq 23\ \mu m$) to maximize the nuclear-resonance-assisted K-shell fluorescence yield.

However, from the experimental data, we obtain $\Sigma_K^{exp} = 0.44(10)\ ph_K/ph_{\Gamma_0}$, which is a factor of 1.9(4) larger than the theoretically predicted value. The $\Sigma_K^{exp}$ value was obtained from a raw experimental count rate $N_K = 0.0162(17)\ ph_K/ph_{\Gamma_0}$ by correcting it for (1) X-ray beam attenuation of $A = 0.44(9)$ along the beam path at the MID beamline and (2) reduced detection efficiency $D_{eff} = 0.084(7)$ in our experiment. The detection efficiency was lower because of a limited solid angle of detection, limited data acquisition time and absorption of K-shell fluorescent photons in the target and in the air.

The large discrepancy indicates possible overvaluation of the magnitudes of the internal conversion coefficients $\alpha$ and $\alpha_K$ in equations (5) and (6). One of the reasons for the overvaluation is that the value of $\omega_K = 0.144(4)$ applied in refs. 51,53 to determine $\alpha_K$ was substantially smaller than the presently accepted value of $\omega_K = 0.19(1)$ (refs. 33,65).

The internal conversion theory (https://bricc.anu.edu.au/)[38] provides smaller values for the internal conversion coefficients for the 12.4-keV to ground-state transition in $^{45}Sc$: $\alpha = 424$, $\alpha_K = 363$ and $\alpha_K/(1+\alpha) = 0.854$. Using these values results in larger values of the resonant cross-section, $\sigma_0 = 1.9 \times 10^{-20}\ cm^2$, and of the partial nuclear-resonance-assisted K-shell fluorescence yield, $\Sigma_K = 0.39\ ph_K/ph_{\Gamma_0}$. The latter value is in good agreement with the experimental value of $\Sigma_K^{exp} = 0.44(10)\ ph_K/ph_{\Gamma_0}$.

Although the number of the detected nuclear decay events was relatively small ($N_d \simeq 93$), the number of $^{45}Sc$ nuclei excited in the experiment was estimated to be much larger amounting to $N_d(1+\alpha)/(\alpha_K\omega_K D_{eff}) \simeq 7 \times 10^3$.

## Nuclear forward scattering

The time dependence of NFS $F(t)$ in the limit of an optically thin nuclear-resonant target irradiated with a broadband pulsed X-ray source at $t = 0$ is given by[11,66]

$$F(t) = F(0)\exp(-\Gamma_\xi t/\hbar), \tag{7}$$

$$F(0) = 4F_{\Gamma_0}\ \xi^2\ \frac{\Gamma_0}{\Gamma_\xi}\exp(-L/L_0), \tag{8}$$

where

$$\xi = \sigma_0 N L f_{LM}/4, \quad \Gamma_\xi = \Gamma + \xi\Gamma_0, \quad \Gamma = \Gamma_0 + \Delta\Gamma, \tag{9}$$

and $f_{LM}$ is the Lamb–Mössbauer factor—the probability of recoil-free elastic nuclear-resonant absorption and emission (Mössbauer effect). In Sc metal, it is rather high, with $f_{LM} \simeq 0.73$ at room temperature $T \simeq 300\ K$, and even at $T \simeq 1{,}000\ K$ it is substantial: $f_{LM} \simeq 0.35$. These numbers are calculated using the Debye model assuming a Debye temperature of $\Theta_D = 355\ K$ for Sc metal[67].

To first order, the Mössbauer effect leads to an unbroadened nuclear resonance line despite atomic vibrations in crystals. However, the second-order Doppler effect can lead both to a temperature-dependent red shift[68,69] and to broadening[70] of the nuclear resonance line. Typically such thermal broadening is negligible for standard Mössbauer resonances such as that of $^{57}Fe$. However, it is not negligible for the $^{45}Sc$ resonance. According to the theory[70], the thermal broadening of the $^{45}Sc$ line should be $\Delta\Gamma_{th} \simeq 63\Gamma_0$ at $T \simeq 2\Theta_D$; $\Delta\Gamma_{th} \simeq 27\Gamma_0$ at $T \simeq \Theta_D/3$; and $\Delta\Gamma_{th} \simeq 0.7\Gamma_0$ at $T \simeq \Theta_D/5$. Only at $T \simeq \Theta_D/10$ does the broadening become $\Delta\Gamma_{th} \simeq 0.005\ \Gamma_0$, that is, much less than the natural linewidth. In our experiment, the incident beam heats the target to a temperature much higher than room temperature, most probably resulting in $\Delta\Gamma_{th} > 50\Gamma_0$. Many different mechanisms could lead to resonance broadening; however, thermal broadening is unavoidable even in perfectly homogeneous targets, unless they are cooled to $T \simeq \Theta_D/10$, which corresponds to about 35 K for a Sc-metal target.

To determine whether NFS photons can be detected in resonance excitation with an XFEL, we estimate the NFS peak count rate $F(0)$ in our experiment using equations (7)–(9). We assume that in equation (1) the incident spectral flux density on the target is $F_{\Gamma_0} \simeq 0.3\ ph\ s^{-1}\Gamma_0^{-1}$; (2) the target sample temperature is high, with $T \gtrsim 2\Theta_D$, resulting in $f_{LM} \simeq 0.35$ and a thermal resonance broadening $\Delta\Gamma_{th} > 50\Gamma_0$; (3) the nuclear target thickness $L = 25\ \mu m$ and therefore the thickness parameter $\xi = 0.16$ (with $\sigma_0 = 1.87 \times 10^{-20}\ cm^2$; see Table 1); and (4) an attenuation factor in the nuclear target $\exp(-L/L_0) = 0.66$ As a result, we obtain for the peak NFS count rate a very small value of $F(0) = 0.0004\ ph\ s^{-1}$. We also note that the NFS signal decays rapidly, with a decay time constant of less than $\tau_0/50$.

If we repeat this calculation for a lower temperature $T \simeq \Theta_D/10$, at which $\Delta\Gamma_{th} \simeq 0$ and $f_{LM} = 0.9$, and choose a target thickness $L = 2L_0 = 120\ \mu m$ optimized for NFS[11] yielding $\xi = 2.0$ and $\exp(-L/L_0) = 0.14$, with these values we would obtain a peak count rate of $F(0) = 0.23\ ph\ s^{-1}$ that is 600 times higher than for $T \gtrsim 2\Theta_D$ (assuming there is no other broadening mechanism than the thermal broadening).

This analysis explains why NFS was not detected in our experiment, which we had optimized for the observation of fluorescent photons following internal conversion. It also shows that in the case of negligible inhomogeneous broadening, a marked improvement in the NFS signal is expected if the nuclear target is cooled to $T \simeq \Theta_D/10$ and the target thickness is optimized for NFS.

A relatively small inhomogeneous broadening $\Delta\Gamma \lesssim 500\Gamma_0$ is crucial for the observation of NFS. The inhomogeneous broadening can be suppressed by using solid-state magnetic nuclear resonance techniques[71–73]

generalized for the nuclear gamma-resonance case[74–76] or by time reversal triggered by magnetic field inversion[77].

## Absolute resonance energy

Absolute values of the wavelengths $\lambda$ and energies $E = hc/\lambda$ of X-ray photons are usually measured in terms of the accurately known Si crystal lattice parameter $a_{Si}$ (refs. 78,79) in X-ray Bragg diffraction experiments on high-purity flawless Si single crystals. In this study, we use the crystal lattice parameter $a_{Si} = 543.101990(1)$ pm of Si at 22.5 °C and the linear thermal expansion coefficient $\gamma = 2.581(2) \times 10^{-6}$ K$^{-1}$ (ref. 79). The $\lambda$-to-$a_{Si}$ relationship is established using Bragg's law as

$$2d_H\sin\theta = \lambda(1 + w_H), \quad w_H = w_H^{(s)}\frac{b-1}{b}, \tag{10}$$

where $\theta$ is the glancing angle of incidence between the wavevector $\mathbf{K}_0$ of the incoming X-rays and the diffracting atomic planes with diffraction vector $\mathbf{H} = (h\,k\,l)$ and interplanar distance $d_H = a_{Si}/\sqrt{h^2 + k^2 + l^2}$ at the Bragg reflection peak. Equation (10) is a generalized Bragg's law[80]. It takes into account a small but potentially important refraction correction $w_H$ in the crystal in a general case of a scattering geometry with a nonzero asymmetry angle $\eta$ between the crystal normal $\hat{\mathbf{z}}$ and $\mathbf{H}$ that is characterized by the asymmetry factor $b = -\sin(\theta + \eta)/\sin(\theta - \eta)$ (Extended Data Fig. 1). The value of $w_H$ is determined through its value $w_H^{(s)}$ in the symmetric scattering geometry with $\eta = 0$ ($b = -1$). To relate $\lambda$ to $a_{Si}$ the angle $\theta$ has to be accurately measured, which is often not straightforward.

In the present study, we apply Bond's method[37], which suggests a measurement of the small angle $\beta = \beta_1 + \beta_2$ between two Bragg-reflecting positions of the crystal instead of $\theta$, where $\beta_i = \theta_i + \eta_i - \pi/2$ (Extended Data Fig. 1). The smaller the $\beta$, the higher the accuracy of the method.

To this end, we choose two Bragg reflections $\mathbf{H}_i = (8\,0\,0)$ and $\mathbf{H}_2 = (0\,8\,0)$ from a Si crystal slab cut parallel to the (1 1 0) planes with an accuracy of $\Delta \lesssim 0.35$ mrad. In this case, $\eta_1 + \eta_2 = \pi/2$, and $\eta_i \simeq \pi/4$ to an accuracy of $\Delta/2$. For the nominal photon energy $E = 12.40$ keV, the angle $\theta \simeq 47.5°$, which results in a favourably small $\beta \simeq 5°$.

Because $\Delta$ is small, the following parameters can be considered equal for both reflections with high accuracy: $b_H = -23.0(2)$, $(b_H - 1)/b_H = 1.043$ and $w_H = 6.12 \times 10^{-6}$, assuming $w_H^{(s)} = 5.87 \times 10^{-6}$. Under these assumptions, Bragg's equation in our case reduces to

$$\lambda = \frac{hc}{E} = \frac{a_{Si}}{4(1 + w_H)}\sin(\pi/4 + \beta/2). \tag{11}$$

By measuring the angle $\beta$ between the two Bragg-reflecting peak positions of the Si crystal and using equation (11), we determine the absolute peak values of the photon energy distributions delivered by the XFEL and relate them to the corresponding relative energies measured by the SSS. In this way, the relationship between the relative and the absolute energy scales is established.

The spectral resolution $\Delta E_B = \sqrt{2}\,\Delta\Theta_i D_H \simeq 38$ meV of the Bond spectrometer is defined mainly by the angular spread $\Delta\Theta_i \simeq 2.4$ μrad of the XFEL beam and the Bragg reflection DuMond tangent (dispersion) $D_H = 11.4$ meV μrad$^{-1}$. Contributions related to the intrinsic energy width $\Delta E_H = 7.7$ meV and the angular width $\Delta\theta_H = 0.7$ μrad of each Bragg reflection are smaller.

Following this procedure, the energy of the $^{45}$Sc nuclear transition is determined as $E_0 = 12{,}389.59^{+0.15(stat)}_{+0.12(syst)}$ eV. The uncertainties are because of the following main contributions that are larger than the spectrometer resolution $\Delta E_B$: (1) A crystal temperature uncertainty of $23.0 \pm 0.5$ °C, a relative energy error and fit errors add together to a statistical error of $\pm 150$ meV. (2) Angular misalignment of the crystal in the off-diffraction planes results in measuring smaller energies. Crystal alignment errors of $\pm 0.12°$ yaw and $\pm 0.25°$ roll—rotation about

$\hat{\mathbf{z}}$ and $\hat{\mathbf{y}}$ axes, respectively—result in an approximately +120 meV systematic error derived from numerical calculations using SHADOW[81].

## Data analysis

To determine resonance energy and width, the experimental raw counts in the incident and detector energy ROIs (Fig. 3b) were normalized using an X-ray gas intensity monitor of MID, binned in incident energy and subsequently fitted using a Gaussian line profile, taking into account the Poisson uncertainty in the experimental count rate. We found that the fit result depends on the bin size and the shift of the binning grid relative to the candidate resonance energy. Therefore, we repeated this analysis for bin sizes between 0.45 eV and 1.11 eV and bin shifts within $\pm$(bin size)/2 around the resonance candidate. This analysis results in a distribution of resonance energies, which we fitted using a Gaussian function. The centre energy of this Gaussian fit is taken as the $^{45}$Sc resonance energy reported in the text, and the width (1 standard deviation) as its uncertainty. Similarly, the initial fits with different bin sizes and bin shifts result in a distribution of $^{45}$Sc resonance widths. An analogous procedure was applied to this distribution to obtain the $^{45}$Sc resonance width and its uncertainty reported in the text. We note that the apparent side bands at $E_i - E_0 \simeq -3.5$ eV and $E_i - E_0 \simeq 3$ eV are artefacts due to noise counts and binning procedure.

## Data availability

Data recorded for the experiment at the EuXFEL are available at https://in.xfel.eu/metadata/doi/10.22003/XFEL.EU-DATA-003159-00.

## Code availability

The codes used to evaluate the data are available from the corresponding author upon request.

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

**Acknowledgements** We thank G. Smirnov and E. Gerdau for their long-standing encouragement and interest and G. G. Paulus for his interest and support. Work at Argonne National Laboratory (ANL) was supported by the U.S. Department of Energy, Office of Science, Office of Basic Energy Sciences, under contract DE-AC02-06CH11357. Work at Texas A&M University and ANL was also supported by the National Science Foundation (grant no. PHY-2012194 'Quantum Optics with Ultra-Narrow Gamma Resonances'). We acknowledge European XFEL in Schenefeld, Germany, for the provision of XFEL laser beamtime at the Materials Imaging and Dynamics instrument located at the SASE-2 beamline and would like to thank the staff for their assistance, in particular J. Wrigley, A. Parenti and S. Karabekyan.

**Author contributions** Y.S. conceived the experiment and supervised the project; R.R. wrote beamtime proposals with input from research groups; Y.S., R.R., O.K., J.E., A. Madsen and G.A.G. coordinated the effort of the research groups; Y.S., D.S., P.L., I.S., A.Z. and J.H designed, constructed, commissioned and provided the $^{45}$Sc motion and detection unit; Y.S., B.S. and P.L. designed and manufactured $^{45}$Sc targets; Y.S., A. Miceli, R.R., I.U., A.Z., J.H. and O.L. selected, tested and provided the X-ray detectors; I.U., R.R., B.M.-G., R.L. and W.H. constructed, tested and provided the Bond spectrometer; G.A.G., C.G., M. Guetg, V.K., N.K., S.L. and W.Q. provided the self-seeded beam with photon-energy scan capabilities; Y.S., J.E., M. Gerharz, R.L., J.H., A.Z., U.B., A.R.-F., J.M., W.J. and M.Y. designed the procedures and carried out data acquisition; Y.S., R.R., J.E., G.A.G., A. Madsen, R.L., W.H., O.L., M. Gerharz, S.L., A.Z., J.H., H.-C.W., N.K., C.G., V.K., W.Q., T.K., U.B., A.R.-F., J.M., W.J. and M.Y. carried out the experiment; J.E., M. Gerharz, R.L., W.H. and X.Z. evaluated and analysed the data; Y.S., O.K., J.E. and X.Z. provided theoretical interpretation; Y.S. and O.K. wrote the first draft of the paper; R.R. and J.E. co-wrote the paper; A. Madsen, G.A.G., P.L., J.H., U.B., M.Y., X.Z. and H.-C.W. discussed and improved the paper.

**Competing interests** The authors declare no competing interests.

**Additional information**
**Correspondence and requests for materials** should be addressed to Yuri Shvyd'ko.

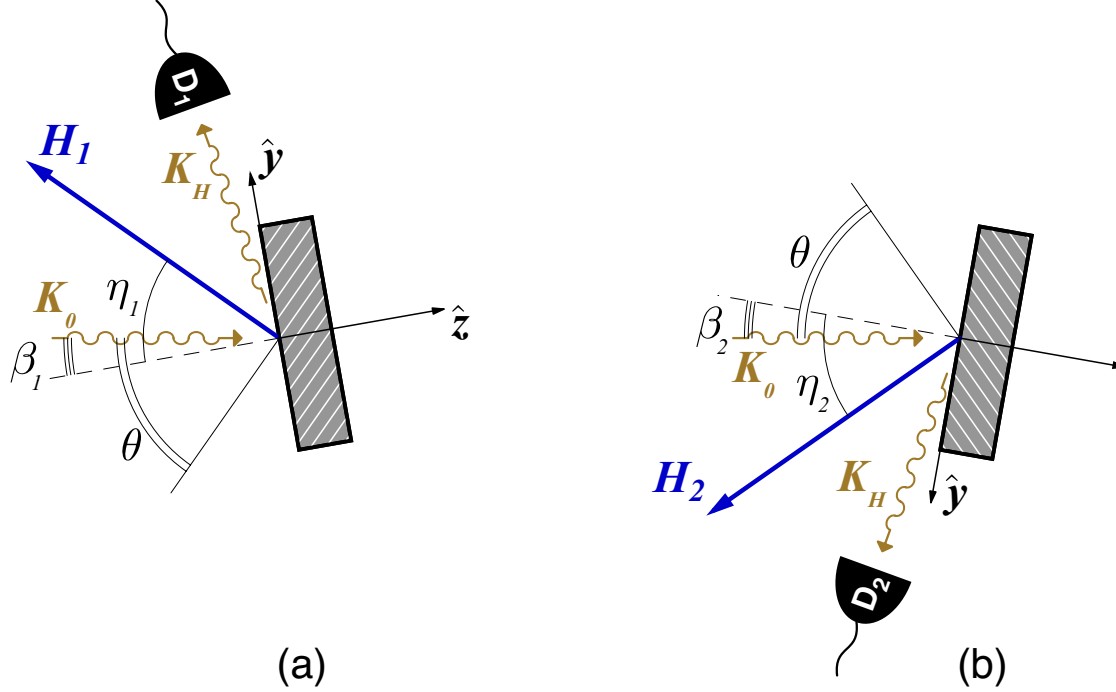

(a)

(b)

**Extended Data Fig. 1 | Schematic of Bragg diffraction of X-rays with wavevector $K_0$ from atomic planes with diffraction vectors (a) $H_1 = (8\ 0\ 0)$ and (b) $H_2 = (0\ 8\ 0)$ from a Si crystal in the Bond spectrometer.** The crystal is rotated by $\beta = \beta_1 + \beta_2$ between the angular positions of the $(8\ 0\ 0)$ and the $(0\ 8\ 0)$ Bragg reflections. Crystal normal $\hat{z} = -(1\ 1\ 0)$. The wavevector of the diffracted X-rays is $K_H$.