## [Peer Review File · Nature]

Manuscript Title: Resonant X-ray excitation of the nuclear clock isomer ^{45}Sc

Reviewer Comments & Author Rebuttals

Reviewer Reports on the Initial Version:

Referees' comments:

Referee #1 (Remarks to the Author):

Congratulations to the team for achieving such beautiful work. The low-lying and long-lived isomeric state of ^{45}Sc was discovered in 1964 [Phys. Rev. 134, B976 (1964)] and is notorious for its high internal conversion coefficient about 400, narrow linewidth about 1eV, the absence of a parent radioactive nucleus, and the lack of efficient x-ray source for pumping. The team has overcome many difficult experimental challenges and the low yield rate of fluorescence by using self-seeded XFEL in EuroXFEL to excite ^{45}Sc 12.4keV nuclear isomeric transition. The delayed K-shell x-ray fluorescence off excited ^{45}Sc is successfully detected. The isomeric transition energy is further pinpointed with an uncertainty about 0.3eV, and many parameters of ^{45}Sc are also measured. The work also manifests that modern XFEL provides a new possibility of exciting even more species of nuclei than the typical workhorse of ^{57}Fe [Nat. Phys. 14, 261 (2018)] and beyond those by 3rd-generation synchrotron radiation. As pointed out by the team, promising applications will be the nuclear clock and ultra-high-precision spectroscopy. This is definitely a milestone in the long scientific journey starting from the discovery of Mössbauer effect back to 1958 and along S. Ruby's pioneering proposal in 1974 of using synchrotron radiation to directly excite nuclei. I therefore strongly recommend the manuscript for publication in Nature. Just a few minor questions for authors' consideration to their future version of the manuscript:

- 1) Shall a short review about other cases of exciting isomer transitions be introduced, e.g., ^{178}Hf , ^{180}Ta , or ^{103}Rh [Hyperfine Interact 167, 833] so on?
- 2) If I understand correctly, ^{45}Sc target is moving back and forth between the XFEL beam line and detectors with pretty high speeds. Shall the relativistic Doppler shift play any role?
- 3) In case the discrepancy between experimental and theoretical yield rate of K-shell X-ray fluorescence is significant, does it imply any corrections to the IC theory?
- 4) Between the vertical red lines but outside ROI around the resonant energy in Fig.3, the noise signal is significantly higher than those in other regions. Are they really just noises?
- 5) There seem to be sidebands around $E_i - E_0 = -3.5\text{eV}$ and 2.8eV in Fig.4. Can the authors comment on the sidebands?
- 6) Can the authors comment on the issue does the pulse area of XFEL, namely the x-ray coherence, play any role?
- 7) Even though the present result is one of the strong motivations for building XFEL. There must be even stronger scientific cases to maintain an accelerator-based nuclear clock. Can the authors comment on this.

In summary, I strongly recommend the manuscript for publication in Nature.

Referee #2 (Remarks to the Author):

This paper accurately determines the resonant energy for generating narrowband light through nuclear resonance excitation. To achieve this, high-intensity XFEL (X-ray Free-Electron Laser) was utilized, and precise measurements were successfully carried out using a simple experimental setup. The high precision achieved opens up prospects for the realization of atomic clocks. The positioning of the objective, which includes this prospect, bears similarities to the study by Masuda et al., published in Nature, Vol. 573, p. 238. However, the methodology differs in terms of the use of high-intensity synchrotron radiation and XFEL as light sources.

Indeed, the utilized nuclear species also differ in the two studies. I believe this research paper asserts its impact in terms of the novelty of the measurement method. Its innovativeness is acknowledged. As one suggestion, it may be worthwhile to cite Masuda et al.'s paper in the references and discuss the differences and advantages. Based on that, it is considered advisable to publish this research paper.

Referee #3 (Remarks to the Author):

The manuscript presents the first excitation of the long-lived nuclear isomer in the ^{45}Sc isotope using x-rays generated by EuXFEL. The excitation of the nucleus was detected by moving the Sc metal foil target out of the way of the x-rays to a position between two off-axis Si detectors. Here the emission of fluorescent photons $K\alpha$ (4.09 keV) and $K\beta$ (4.46 keV) was recorded. The 12.4 keV photons could not be detected. The excitation energy of this isomer was improved by two orders of magnitude. The presented data was acquired in 30 h. These results present an important step for the possible development of a nuclear clock with ^{45}Sc . There has been significant progress in the development of nuclear clocks, most of which concerns the isomer in ^{229}Th . The possibility of using ^{45}Sc has until now not been widely considered as a candidate, which is evident from the list of references provided in this manuscript. This work can potentially change this.

The timing of this manuscript is ideal, given the recent progress in the field of nuclear clocks, with the most recent progress reported in May (<https://doi.org/10.1038/s41586-023-05894-z>). The presented results mark a milestone in establishing a nuclear clock using ^{45}Sc . The possible application of such a clock is motivated.

The methodology and results are clearly outlined and discussed. Information is provided about the properties of the target material, x-rays pulses and the procedure used to obtain the data plotted in the figures. The presented data are of high quality. The figures are well presented and provide the necessary information to understand and interpret the results. The absence data for nuclear forward scattering is clearly explained and the conclusions are supported by quantitative assessment as well, which is added in the Methods section.

For the data analysis the tools appropriate for low statistics were employed. In the Methods: Data analysis, the final step for obtaining the uncertainties could be further clarified: how were the results of the Gaussian fits for different bin sizes combined into the reported results? Based on the current phrasing, it is not clear what procedure was used. Possible systemic uncertainties were considered where relevant.

The conclusions of the manuscript are sound and are based on firm results. Perhaps the feasibility of the future application of a Sc based nuclear clock could include more (quantitative) information on the requirements. This would enable the readers to assess when and how could such a clock be realized.

Below is a short list of questions which could be helpful to provide further relevant information to the readers:

1. Are there any other promising isomers which can be probed using the current setup?
2. How do the nuclear electromagnetic moments affect the applicability of this transition in fundamental tests?
3. ^{229}Th is also highlighted due to the high Lamb-Mössbauer factor at room temperature. Why was the Sc metal foil target chosen for this experiment and are there any obstacles in the way of using ^{229}Th in the next experiments?
 - a) Would such a target be able to endure the irradiation?
 - b) Perhaps a reference could be added at the end of the second to last sentence on Page 2 paragraph 1.
 - c) To what extent do the thermal properties of the Sc metal foil limit this experiment? Would focusing the beam onto the target significantly improve the results?

Small comments:

- Affiliation number 3: I believe there is a mistake in the address of GSI.
- Table 1: for some of the properties the uncertainties are not indicated. The uncertainties should be added, or their absence should be motivated.
- There is a re-evaluated value for the nuclear magnetic moment of the ground state of ^{45}Sc (<https://doi.org/10.1016/j.cplett.2016.08.002>) in case the authors wish to find the most recent values.
- What is the tuning range of the EuXFEL? This could be added to the end of the 1st sentence on page 2, paragraph 1.
- Page 1 2nd paragraph: I suggest removing the brackets around "due to the tiny size of nuclei" and add that the small nuclear electromagnetic moments (presented in Table 1) also play an important role.
- It is mentioned that the irradiation source was defocused to reduce the possibility of damage. What was the beam size on the Sc target?
- It could be interesting to add to the first paragraph of the section *Measurements and results* the number of Sc atoms which were excited to obtain the data. The reader might be able to calculate this based on the numbers provided in the methods, but it would be informative to add this to the main text. For example, a sentence specifying the number of x-rays (already in the text), number of excited Sc nuclei and the recorded count rate during the 30h measurement would be useful.
- Page 4 paragraph 2: It would be useful to quantify the statistical significance of the data points shown in Figure 3c. Is the rate in this zoomed-in region consistent with the background rate in other energy regions?

While the advantages of a nuclear clock using ^{45}Sc are clearly described, it is not evident how such a clock could be realized. As a non-expert, it is not immediately apparent how far we are from such a clock – are huge technological breakthroughs still required? Or is the path now clear. The text does address the need and progress in higher-energy frequency combs, but does not provide an indication of what is required on other fronts. My recommendation is to add more information regarding e.g. the proposed target material, quantify the properties of the x-ray pulses needed for the clock and procedure for establishing a nuclear clock. This will make it easier for the general readership of Nature to appreciate the progress towards a scandium clock, and the importance of this impressive measurement. This will elevate the impact of the manuscript and clarify the feasibility of this method in the future.

The recently published ^{229}Th manuscript is not cited in the current manuscript, probably since this work has been finalized before the ^{229}Th paper was published. In any case, those results don't change the relevance or value of the current manuscript, on the contrary, it highlights the increasing interest and progress on this front. It seems appropriate to refer to <https://doi.org/10.1038/s41586-021-03276-x>, Coherent X-ray–optical control of nuclear excitons.

The abstract is clear, and it gives enough context to the reader to understand the topic and the

main results presented in the manuscript. The same is true for the introduction and the conclusions. As mentioned above, the paragraph in the conclusions discussing the requirements for the nuclear clock could include more quantitative assessment and elaborate on how such a Sc clock could operate.

Author Rebuttals to Initial Comments:

Dear reviewers,

thank you very much for recommending our manuscript for publication in Nature and for your valuable comments. Please see below point-to-point response to your questions and comments.

We are also appending to this document the revised manuscript in the same format as the originally submitted manuscript with changes highlighted for your convenience in blue.

With best regards.

Yuri Shvyd'ko

on behalf of co-authors

Referee #1 (Remarks to the Author):

Congratulations to the team for achieving such beautiful work. The low-lying and long-lived isomeric state of ^{45}Sc was discovered in 1964 [Phys. Rev. 134, B976 (1964)] and is notorious for its high internal conversion coefficient about 400, narrow linewidth about 1eV, the absence of a parent radioactive nucleus, and the lack of efficient x-ray source for pumping. The team has overcome many difficult experimental challenges and the low yield rate of fluorescence by using self-seeded XFEL in EuroXFEL to excite ^{45}Sc 12.4keV nuclear isomeric transition. The delayed K-shell x-ray fluorescence off excited ^{45}Sc is successfully detected. The isomeric transition energy is further pinpointed with an uncertainty about 0.3eV, and many parameters of ^{45}Sc are also measured. The work also manifests that modern XFEL provides a new possibility of exciting even more species of nuclei than the typical workhorse of ^{57}Fe [Nat. Phys. 14, 261 (2018)] and beyond those by 3rd-generation synchrotron radiation. As pointed out by the team, promising applications will be the nuclear clock and ultra-high-precision spectroscopy. This is definitely a milestone in the long scientific journey starting from the discovery of Mssbauer effect back to 1958 and along S. Ruby's pioneering proposal in 1974 of using synchrotron radiation to directly excite nuclei. I therefore strongly recommend the manuscript for publication in Nature. Just a few minor questions for authors' consideration to their future version of the manuscript:

1) Shall a short review about other cases of exciting isomer transitions be introduced, e.g., ^{178}Hf , ^{180}Ta , or ^{103}Rh [Hyperfine Interact 167, 833] so on?

= Thank you very much for the reference. We have added it to the paper together with the following comment: "Other approaches have employed bremsstrahlung to populate indirectly long-lived isomeric states like the 39-keV level of ^{103}Rh [Reference] via higher-energy broadband excited states".

2) If I understand correctly, ^{45}Sc target is moving back and forth between the XFEL beam line and detectors with pretty high speeds. Shall the relativistic Doppler shift play any role?

= The answer is no, because (1) we have a broadband spectrum of incident x-rays, (2) the target is at rest when the nuclei are irradiated or decaying, and (3) another reference resonance absorber would be required to detect the Doppler shift.

3) In case the discrepancy between experimental and theoretical yield rate of K-shell X-ray fluorescence is significant, does it imply any corrections to the IC theory?

= The answer is no. The correction is in agreement with the modern IC theory [23,24].

4) Between the vertical red lines but outside ROI around the resonant energy in Fig.3, the noise signal is significantly higher than those in other regions. Are they really just noises?

= Yes, this is noise. The noise level is higher in the region between the red lines, because the energy scan range was reduced to this smaller region after the candidate resonance energy was located. We write in the paper on p.3: "Because of this scan-range reduction, the density of the detected (background) photons appears to be larger between the vertical red lines."

5) There seem to be sidebands around $E_i - E_0 = -3.5\text{eV}$ and 2.8eV in Fig.4. Can the authors comment on the sidebands?

= Thank you very much for this comment. These are artifacts caused by the noise counts at these energies and binning procedure. We added to Methods section Data analysis the following sentence: "We note that the apparent side bands at $E_i - E_0 \simeq -3.5\text{ eV}$ and $E_i - E_0 \simeq 2\text{ eV}$ are artifacts due to noise counts and binning procedure."

6) Can the authors comment on the issue does the pulse area of XFEL, namely the x-ray coherence, play any role?

= In this experiment coherence didn't play any role. As we state in the paper, the record high value of the spectral flux was essential to the choice of this x-ray source.

7) Even though the present result is one of the strong motivations for building XFEL. There must be even stronger scientific cases to maintain an accelerator-based nuclear clock. Can the authors comment on this.

= As we discuss in Background and Objectives and in Discussion sections, motivations for further studies using Sc-resonance include ultra-high-precision spectroscopy, extreme metrology in the regime of hard X-rays, along with nuclear clock technology, and other similar applications.

In summary, I strongly recommend the manuscript for publication in Nature.

= Thank you very much for your very useful comments and recommendations.

Referee #2 (Remarks to the Author):

This paper accurately determines the resonant energy for generating narrowband light through nuclear resonance excitation. To achieve this, high-intensity XFEL (X-ray Free-Electron Laser) was utilized, and precise measurements were successfully carried out using a simple experimental setup. The high precision achieved opens up prospects for the realization of atomic clocks. The positioning of the objective, which includes this prospect, bears similarities to the study by Masuda et al., published in Nature, Vol. 573, p. 238. However, the methodology differs in terms of the use of high-intensity synchrotron radiation and XFEL as light sources. Indeed, the utilized nuclear species also differ in the two studies. I believe this research paper asserts its impact in terms of the novelty of the measurement method. Its innovativeness is acknowledged. As one suggestion, it may be worthwhile to cite Masuda et al.'s paper in the references and discuss the differences and advantages. Based on that, it is considered advisable to publish this research paper.

= Thank you very much for the recommendation and for the suggestion. The reference to the

paper by Masuda et al. is included into abstract and introduction with the following sentence added on page 1: “An alternative, though indirect route to populate the isomeric state could proceed via X-ray excitation of the 29.2 keV broadband second excited state of ^{229}Th [3].”

Referee #3 (Remarks to the Author): The manuscript presents the first excitation of the long-lived nuclear isomer in the ^{45}Sc isotope using x-rays generated by EuXFEL. The excitation of the nucleus was detected by moving the Sc metal foil target out of the way of the x-rays to a position between two off-axis Si detectors. Here the emission of fluorescent photons K_α (4.09 keV) and K_β (4.46 keV) was recorded. The 12.4 keV photons could not be detected. The excitation energy of this isomer was improved by two orders of magnitude. The presented data was acquired in 30 h. These results present an important step for the possible development of a nuclear clock with ^{45}Sc . There has been significant progress in the development of nuclear clocks, most of which concerns the isomer in ^{229}Th . The possibility of using ^{45}Sc has until now not been widely considered as a candidate, which is evident from the list of references provided in this manuscript. This work can potentially change this.

The timing of this manuscript is ideal, given the recent progress in the field of nuclear clocks, with the most recent progress reported in May (<https://doi.org/10.1038/s41586-023-05894-z>). The presented results mark a milestone in establishing a nuclear clock using ^{45}Sc . The possible application of such a clock is motivated.

= Thank you very much for the reference. It is included into our manuscript in the abstract and in the introduction.

The methodology and results are clearly outlined and discussed. Information is provided about the properties of the target material, x-rays pulses and the procedure used to obtain the data plotted in the figures. The presented data are of high quality. The figures are well presented and provide the necessary information to understand and interpret the results. The absence data for nuclear forward scattering is clearly explained and the conclusions are supported by quantitative assessment as well, which is added in the Methods section.

For the data analysis the tools appropriate for low statistics were employed. In the Methods: Data analysis, the final step for obtaining the uncertainties could be further clarified: how were the results of the Gaussian fits for different bin sizes combined into the reported results? Based on the current phrasing, it is not clear what procedure was used. Possible systemic uncertainties were considered where relevant.

= In order to avoid any bias in binning the data, we repeated the fit of the experimental data with a large number of different bin sizes and shifts of the binning grid around the candidate resonance energy. These fits result in a distribution of resonance energies (one per fit). We then created a histogram from this distribution, and fitted it with a Gaussian. The center energy of this final Gaussian fit is then taken as the resonance energy reported in the manuscript, and its width (one standard deviation) as the uncertainty of the resonance energy. Next to a distribution of resonance energies, the analysis with different bin sizes/shifts also results in a distribution of resonance widths. Using an analogous analysis (histogram + Gaussian fit), the resonance width and its uncertainty are evaluated.

In response to this remark, we improved the description of this procedure in the methods section “Data analysis” along the lines of the above discussion.

The conclusions of the manuscript are sound and are based on firm results. Perhaps the feasibility of the future application of a Sc based nuclear clock could include more (quantitative) information on the requirements. This would enable the readers to assess when and how could such a clock be realized.

Below is a short list of questions which could be helpful to provide further relevant information to the readers:

1. Are there any other promising isomers which can be probed using the current setup?

= As we are explaining in the manuscript, there are other long-lived transitions, but unfortunately with much higher excitation energy - like ^{109}Ag [11,16] - inaccessible yet by the present accelerator-based x-ray sources.

2. How do the nuclear electromagnetic moments affect the applicability of this transition in fundamental tests?

= They can be useful as well as detrimental. The impact of the nuclear magnetic moments will start to become critical in the next step of observation of NFS in ^{45}Sc .

3. Sc_2O_3 is also highlighted due to the high Lamb-Mössbauer factor at room temperature. Why was the Sc metal foil target chosen for this experiment and are there any obstacles in the way of using Sc_2O_3 in the next experiments?

a) Would such a target be able to endure the irradiation?

b) Perhaps a reference could be added at the end of the second to last sentence on Page 2 paragraph 1.

c) To what extent do the thermal properties of the Sc metal foil limit this experiment? Would focusing the beam onto the target significantly improve the results?

= For the experiment presented in the current paper, there was practically no difference whether using Sc metal, Sc_2O_3 , or any other Sc-containing material with high nuclear-resonance optical thickness and x-ray transparency. To observe NFS, which we are planning in the next step, the choice of Sc metal, or Sc_2O_3 will be more delicate. We will definitively discuss this topic in more details in our next publication. Here in the revised version of the manuscript, we have added the following additional information to Methods section Experimental challenges:

“The choice of the ^{45}Sc target was dictated by the necessity of achieving high nuclear resonance optical thickness, low absorption of the nuclear decay products, resilience to radiation damage, simplicity in fabrication and integration into the motion and detection system.”

“(the beam footprint was about 2 mm^2)”

Small comments:

• Affiliation number 3: I believe there is a mistake in the address of GSI.

= Thank you for pointing this out, we have corrected this

• Table 1: for some of the properties the uncertainties are not indicated. The uncertainties should be added, or their absence should be motivated.

= Uncertainties have been added.

- There is a re-evaluated value for the nuclear magnetic moment of the ground state of ^{45}Sc (<https://doi.org/10.1016/j.cplett.2016.08.002>) in case the authors wish to find the most recent values.

= Thank you very much for the alternative value of the magnetic moment.

- What is the tuning range of the EuXFEL? This could be added to the end of the 1st sentence on page 2, paragraph 1.

= We added the tuning range for the European XFEL in seeded mode to Methods section Experimental challenges: "EuXFEL currently delivers photons in self-seeding mode in the energy range of 6 – 14 keV".

- Page 1 2nd paragraph: I suggest removing the brackets around due to the tiny size of nuclei and add that the small nuclear electromagnetic moments (presented in Table 1) also play an important role.

= Brackets were removed and the following was added: "and small magnitudes of nuclear electromagnetic moment".

- It is mentioned that the irradiation source was defocused to reduce the possibility of damage. What was the beam size on the Sc target?

= The following was added to Methods section Experimental challenges: "(the beam footprint was about 2 mm^2)"

- It could be interesting to add to the first paragraph of the section Measurements and results the number of Sc atoms which were excited to obtain the data. The reader might be able to calculate this based on the numbers provided in the methods, but it would be informative to add this to the main text. For example, a sentence specifying the number of x-rays (already in the text), number of excited Sc nuclei and the recorded count rate during the 30h measurement would be useful.

= We added a footnote at the beginning of the Measurements and Results section: "While the number of the detected nuclear decay events was relatively small ($\simeq 93$) the number of ^{45}Sc nuclei excited in the experiment was estimated to be much larger $\simeq 7 \times 10^3$ as explained in Methods."

We also added more details in the Methods section Incoherent scattering count rates: "While the number of the detected nuclear decay events was $N_d \simeq 93$, i.e., was relatively small, the number of ^{45}Sc nuclei excited in the experiment was estimated to be much larger amounting to $N_d(1 + \alpha)/\alpha_K \omega_K D_{\text{eff}} \simeq 7 \times 10^3$."

- Page 4 paragraph 2: It would be useful to quantify the statistical significance of the data points shown in Figure 3c. Is the rate in this zoomed-in region consistent with the background rate in other energy regions?

= We did not observe any statistically significant effect in this region, consistent with our theoretical analysis of the nuclear forward scattering. To illustrate this, we summed up the raw count data shown in Fig. 3 of the main text and in a $\pm 2\text{eV}$ range around the ^{45}Sc resonance energy along the incidence energy axis. Fig. 1 then shows this raw count data as function of the

Figure 1: Raw counts in a $\pm 2\text{eV}$ incident energy region around the ^{45}Sc resonance, as function of detection energy. The right panel shows a zoom of the left panel in the low-count region.

detector energy, compiled in bins of 250 eV size. In the left panel, the resonance counts around the K_{α} and K_{β} lines can clearly be seen high above the noise floor. The right panel shows a magnification along the y-axis. Clearly, there is no increase in count rate around 12.4 keV which exceeds the individual error bars of the background counts.

- While the advantages of a nuclear clock using ^{45}Sc are clearly described, it is not evident how such a clock could be realized. As a non-expert, it is not immediately apparent how far we are from such a clock are huge technological breakthroughs still required? Or is the path now clear. The text does address the need and progress in higher-energy frequency combs, but does not provide an indication of what is required on other fronts. My recommendation is to add more information regarding e.g. the proposed target material, quantify the properties of the x-ray pulses needed for the clock and procedure for establishing a nuclear clock. This will make it easier for the general readership of Nature to appreciate the progress towards a scandium clock, and the importance of this impressive measurement. This will elevate the impact of the manuscript and clarify the feasibility of this method in the future.

We added the following sentence at the end of the Discussion section: "Specifications for the x-ray source and for the nuclear clock procedure conceptually outlined in [46] will become more explicit after observation of ^{45}Sc NFS and optimization of the ^{45}Sc target for maximum NFS signal strength and smallest linewidth of the actual ^{45}Sc resonance."

The recently published 229Th manuscript is not cited in the current manuscript, probably since this work has been finalized before the 229Th paper was published. In any case, those results don't change the relevance or value of the current manuscript, on the contrary, it highlights the increasing interest and progress on this front. It seems appropriate to refer to <https://doi.org/10.1038/s41586-021-03276-x>, Coherent X-ray-optical control of nuclear excitons.

= The reference to the recently published paper was added as [6] in the revised manuscript. We are not doing any coherent control of the nuclear excitons. Therefore, we don't think the other suggested reference is appropriate in our present manuscript.

The abstract is clear, and it gives enough context to the reader to understand the topic and the main results presented in the manuscript. The same is true for the introduction and the conclusions. As mentioned above, the paragraph in the conclusions discussing the requirements for

the nuclear clock could include more quantitative assessment and elaborate on how such a Sc clock could operate.

= Added to the conclusions sections. See the discussion above.